# Twist operator correlators and isomonodromic tau functions from modular Hamiltonians

**Hewei Frederic Jia**

*Center for Quantum Mathematics and Physics (QMAP)*
*Department of Physics & Astronomy, University of California, Davis, CA 95616 USA*

*E-mail:* fjia@ucdavis.edu

ABSTRACT: We introduce a novel approach for computing the twist operator correlators (TOC) in two-dimensional conformal field theories (2d CFT) and the closely related isomonodromic tau functions. The method stems from the formal path integral representation of the ground state reduced density matrix in 2d CFT, and exploits properties of the associated modular Hamiltonians. For a class of genus-zero TOC/tau functions associated with branched covers with non-abelian monodromy group, we present: i) a determinantal representation derived from the correlation matrix method for free fermions, and ii) a formal integral representation derived from the universal single-interval modular Hamiltonians. For the class of genus-zero TOC/tau functions, we also argue an approximate factorization property, utilizing the known ground state correlation structure of large-$c$ holographic CFT and the universality of genus-zero TOCs. We provide explicit examples for verifying the determinantal representation and the approximate factorization property.

## 1 Introduction, summary of results and discussions

The goal of this paper is to uncover novel representations and properties of the twist operator correlators (TOC) in two-dimensional conformal field theory (2d CFT) and the closely related tau functions of isomonodromic type, utilizing techniques and intuitions from quantum information theory.

We first recall the general path integral definition of TOCs in terms of the monodromy data of branched covers [1, 2], and refer the reader to [3] and references therein for the relevance of TOCs in physical contexts such as symmetric product orbifold and string theory.

**Definition 1.1** (Twist operator correlator). *A monodromy data is a pair* $\mathbf{m} = (\boldsymbol{\sigma}, \boldsymbol{z}) \in S_N^M \times \bar{\mathbb{C}}^M$. *Twist operator correlator/partition function with prescribed monodromy* $\mathbf{m}$ *for a generic 2d CFT $\mathcal{C}$ is defined by path integral for $N$ copies of $\mathcal{C}$ with monodromy conditions for fundamental fields* $\{\varphi_I\}_{I=1,\cdots,N}$ *specified by* $\mathbf{m}$

$$\mathcal{Z}_{\mathbf{m}}(\boldsymbol{z}|\boldsymbol{\sigma}) = \left\langle \prod_i \sigma_i(z_i) \right\rangle := \int_{\varphi_I(\xi_i \circ z) = \varphi_{\sigma_i(I)}(z)} [D\varphi]\, e^{-\sum_I S[\varphi_I]} \tag{1.1}$$

where $\boldsymbol{\xi}$ are generating loops in $\pi_1\left(\mathbb{CP}^1 \setminus \boldsymbol{z}\right)$ and $\xi_i \circ z$ denotes continuation along path $\xi_i$. The monodromy data $\mathbf{m}$ is naturally identified as the monodromy data of a branched cover

$$\phi_{\mathbf{m}} : \Sigma \to \mathbb{CP}^1$$

with branch locus $\boldsymbol{z}$ (i.e., critical values of $\phi_{\mathbf{m}}$) and corresponding permutation monodromies $\boldsymbol{\sigma}$. In other words, the twist operator correlator $\mathcal{Z}_{\mathbf{m}}$ is the partition function of $\mathcal{C}$ on $\Sigma$ evaluated in the conformal frame where base $\mathbb{CP}^1$ has flat metric.[1]

The TOCs in physics literature are closely related to the tau functions of isomonodromic type in math literature. The tau function on Hurwitz space, the moduli space of branched covers, associated with a branched cover $\phi_{\mathbf{m}}$ is studied in [4] and is defined as

$$\partial_{z_i} \log \tau_{\mathbf{m}} := \frac{1}{12} \operatorname{Res}_{z=z_i} S_{\phi_{\mathbf{m}}}(z) \tag{1.2}$$

where $S_{\phi_{\mathbf{m}}}(z)$ is the sum of Bergman projective connections of $\phi_{\mathbf{m}}$ at pre-images of $z$ evaluated in base coordinate $z$ and we refer to [3, 4] for more details. The tau functions on Hurwitz space are special cases of the more general isomonodromic tau functions [5] associated with rank $N$ matrix Fuchsian systems while specializing to quasi-permutation monodromies [6].

We showed in [3] that for generic 2d CFT $\mathcal{C}$ and branched covers of genus zero and one, the following relation holds between TOCs and tau functions on Hurwitz space:

$$\mathcal{Z}_{\mathbf{m}} = \begin{cases} |\tau_{\mathbf{m}}|^{2c} & g = 0 \\ |\tau_{\mathbf{m}}|^{2c}|\eta(\tau_{\mathbf{m}})|^{2c}\mathcal{Z}\left(\tau_{\mathbf{m}}, \bar{\tau}_{\mathbf{m}}\right) & g = 1 \end{cases} \tag{1.3}$$

where $c$ is the central charge of CFT $\mathcal{C}$, $\eta(\tau)$ is Dedekind eta function, $\mathcal{Z}(\tau, \bar{\tau})$ is torus partition function of $\mathcal{C}$, and the period $\tau_{\mathbf{m}} = \tau_{\mathbf{m}}(\boldsymbol{z}|\boldsymbol{\sigma})$ of covering torus is viewed as a function of branch locus $\boldsymbol{z}$. Physically, the universal, only central-charge-dependent part of TOC arises from the Weyl anomaly due to Weyl transformation induced by the covering map. In this paper, we will focus on the genus-zero case where the tau functions are the holomorphic part of the $c = 1$ TOCs.

The relevance of quantum information theory for TOC/tau functions arises from the formal path integral representation of the reduced density matrix of the ground state in 2d CFT. As the representation is quite standard in physics literature, we refer readers unfamiliar with the idea to, e.g., the original paper [2] and the review [7] for more details. In this introduction, we proceed by illustrating with a simple example associated with a branched cover with non-abelian monodromy group.

Consider a four-point TOC/tau function, associated with a degree-three genus-zero branched cover, with the following monodromy data (the meaning of the subscripts in the TOC/tau

---

[1]Technically, a cut-off is required at infinity on $\mathbb{CP}^1$; this gives trivial contribution to the $\boldsymbol{z}$-dependence of the twist operator correlator $\mathcal{Z}_{\mathbf{m}}$ [1].

function will become clear):

$$\mathcal{Z}_{(2,2)}(\boldsymbol{z}) = \left|\tau_{(2,2)}(\boldsymbol{z})\right|^{2c}$$
$$\sigma_1 = \sigma_2 = (12), \quad \sigma_3 = \sigma_4 = (13). \tag{1.4}$$

It follows from the standard path integral representation of ground state reduced density matrix in 2d CFT that the TOC admits the following identification as a density matrix/exponentiated modular Hamiltonian correlator:

$$\mathcal{Z}_{(2,2)}(\boldsymbol{z}) = \text{Tr}\left[\rho_R\left(\rho_{R_1} \otimes \rho_{R_2}\right)\right] = \langle \rho_{R_1}\rho_{R_2}\rangle,$$
$$R = R_1 \cup R_2, \quad R_1 = (z_1, z_2), \quad R_2 = (z_3, z_4), \tag{1.5}$$

where $\langle \cdot \rangle$ denotes ground state expectation value, and the density matrices are understood as those of the intervals $R, R_1, R_2$. The reason behind the identification is as follows. The formal path integral on the three sheets of the branched cover with cuts at $R_1$ and/or $R_2$ are understood as preparing the reduced density matrices $\rho_R, \rho_{R_1}, \rho_{R_2}$, and the trace in $\text{Tr}\left[\rho_R\left(\rho_{R_1} \otimes \rho_{R_2}\right)\right]$ glues the sheets together and leads to the defining path integral for TOC in Definition 1.1. In particular, the first sheet of the branched cover corresponds to $\rho_R$, the second sheet to $\rho_{R_1}$, and the third sheet to $\rho_{R_2}$.

The example above is the $(n_1, n_2) = (2,2)$ case of the class of genus-zero four-point TOC/tau functions studied in this paper:

$$\mathcal{Z}_{(n_1,n_2)} = \left|\tau_{(n_1,n_2)}\right|^{2c} = \text{Tr}\left[\rho_R\left(\rho_{R_1}^{n_1-1} \otimes \rho_{R_2}^{n_2-1}\right)\right] = \left\langle \rho_{R_1}^{n_1-1}\rho_{R_2}^{n_2-1}\right\rangle$$
$$\sigma_1 = \sigma_2^{-1} = (1\cdots n_1), \quad \sigma_3 = \sigma_4^{-1} = (1\ n_1+1\cdots n_1+n_2-1)$$
$$R = R_1 \cup R_2, \quad R_1 = (z_1, z_2), \quad R_2 = (z_3, z_4), \tag{1.6}$$

where the associated monodromy data can be easily read off from the path integral representation of reduced density matrices.

From the perspective of their identifications as density matrix correlators, the universality of the genus-zero TOC/tau functions above stems from their representations as ground state expectation value of single-interval density matrices, and the universality of single-interval modular Hamiltonian $\mathcal{H}_{R_i}$ [8–10]:

$$\rho_{R_i} \propto e^{-\mathcal{H}_{R_i}}, \quad R_i = (a_i, b_i)$$
$$\mathcal{H}_{R_i} = \int_{R_i} dz \frac{(z-a_i)(z-b_i)}{b_i - a_i}T(z) + \frac{c}{6}\log\frac{b_i - a_i}{\epsilon}\mathbb{1} + \text{anti-holo.}, \tag{1.7}$$

where $T(z)$ is the holomorphic stress-tensor of the 2d CFT.

While in general genus-zero TOC/tau functions, including the ones in (1.6), can in principle be calculated from the Liouville action associated with the branched cover [1, 4], or directly from the definition[2]

$$\partial_{z_i} \log \tau_{\mathbf{m}} = \frac{1}{12}\text{Res}_{z=z_i}\sum_I\{\psi_{\mathbf{m}}^I, z\}, \tag{1.8}$$

---

[2]Physically, this is equivalent to the stress-tensor method for TOC [3, 11].

where $\psi_{\mathbf{m}}^I$ are inverses of $\phi_{\mathbf{m}}$ and $\{\cdot, z\}$ denotes Schwarzian derivative with respect to $z$, to perform the calculation one needs the explicit knowledge of how branched cover $\phi_{\mathbf{m}}$ varies under deformation of branch locus $\boldsymbol{z}$, i.e., the explicit solution of an isomonodromic deformation problem. Such direct evaluation of TOC/tau functions is generally obstructed by the difficulty of explicitly constructing branched covers with prescribed monodromy, especially those with non-abelian monodromy group, and previous expressions for TOC/tau functions [1, 4, 12, 13] generally leave implicit the dependence on branch locus $\boldsymbol{z}$ in terms of certain unknown coefficients in the rational function $\phi_{\mathbf{m}}$.

Interestingly, the density matrix representations of TOC/tau functions in (1.6) entirely avoid the need for explicit construction of the associated branched covers, and can be evaluated using techniques initially developed in the contexts of condensed matter/quantum information theory.

In particular, utilizing the universality of the genus-zero TOC/tau functions in (1.6), we may as well evaluate them in simple CFTs such as free fermions where the continuum answers can be extracted from the lattice set-up where the correlation matrix method of [14] applies. The method facilitates the evaluation of the density matrix correlators relevant for our purpose by utilizing the simple relation between modular Hamiltonian kernel/matrix and correlation matrix in Gaussian states. This approach is adopted in §3.

Moreover, the universal continuum modular Hamiltonian expression (1.7) can also in principle be used to compute TOC/tau functions in (1.6). We discuss more details and subtleties on computing TOC/tau functions using the continuum modular Hamiltonians in §5.

**Summary of results**

Our main results are:

- A novel determinantal representation for the class of TOC/tau functions in (1.6). This is derived using the correlation matrix method for free fermions. We claim that the non-trivial part of the four-point tau function $\mathfrak{T}_{(n_1, n_2)}$, defined by:

$$\tau_{(n_1, n_2)}(\boldsymbol{z}) = \mathcal{L}_{(n_1, n_2)}(\boldsymbol{z}) \, \mathfrak{T}_{(n_1, n_2)}(x), \quad x = \frac{z_{12} z_{34}}{z_{13} z_{24}}, \tag{1.9}$$

where the leg factor $\mathcal{L}_{(n_1, n_2)}(\boldsymbol{z})$ is defined in §2.1, admits the following determinantal representation in terms of free fermion correlation matrices (claim 3.1):

$$\left| \mathfrak{T}_{(n_1, n_2)}(x) \right|^2 = \lim_{\substack{l_1, l_2, d \to \infty \\ x(l_1, l_2, d) \text{ fixed}}} \mathcal{L}_{(n_1, n_2)}^{-2}(l_1, l_2, d) \det \left( \mathsf{M}_{R_1, R_2}^{(n_1, n_2)} \right) \tag{1.10}$$

where the equality is understood to hold up to $x$-independent overall constant. $l_1, l_2$ are the sizes of the intervals $R_1, R_2$ on the lattice and $d$ their separation, $x(l_1, l_2, d)$ and $\mathcal{L}_{(n_1, n_2)}(l_1, l_2, d)$ are the appropriate cross-ratio and leg factor on lattice, and the

$l_1 + l_2 + 2$-dimensional square matrix $\mathsf{M}^{(n_1,n_2)}_{R_1,R_2}$ is defined in terms of correlation matrices of free fermions; see §3.2 for more details. In §4, we verify the claim in the $(n_1, n_2) = (2, 2)$ case where the exact tau function $\mathcal{T}_{(2,2)}(x)$ is known, and find very good agreement between the exact expression and continuum limit of lattice data. We also present a multi-interval generalization of the determinantal representation in claim 3.2.

- An approximate factorization property for the class of TOC/tau functions in (1.6):

> **Claim.** *There exists $x^* \in \left(\frac{1}{2}, 1\right)$ such that $\forall x \in (0, x^*)$,*
>
> $$\left| \frac{\mathcal{T}_{(n_1,n_2)}(x)}{\mathcal{T}^{\text{fac.}}_{(n_1,n_2)}(x)} \right| = 1 + \epsilon, \quad |\epsilon| \ll 1. \tag{1.11}$$

where $\mathcal{T}^{\text{fac.}}_{(n_1,n_2)}(x)$ is associated with the factorized TOC/tau function; see §2.2 for more details. This is argued based on the known correlation structure of the ground state of large-$c$ holographic CFT and the universality of the genus-zero TOCs. A priori, while the TOCs generally factorize in the $x \to 0$ limit, there is no reason to expect them to approximately factorize in a finite range of cross-ratios as the associated branched cover is connected. In §4, we verify the claim in the $(n_1, n_2) = (2, 2)$ case where the exact tau function $\mathcal{T}_{(2,2)}(x)$ is known, and present more evidence in other two-interval examples by comparing with lattice data.

We also study the class of TOC/tau function in (1.6) using the continuum modular Hamiltonian (1.7). This results in a formal integral representation for TOC/tau functions in terms of integrated stress-tensor correlators:

$$\frac{\mathcal{T}_{(n_1,n_2)}(x)}{\mathcal{T}^{\text{fac.}}_{(n_1,n_2)}(x)} = \prod_{\substack{l_1+l_2>1 \\ l_1,l_2 \neq 0}} \exp\left\{ \frac{\alpha_1^{l_1} \alpha_2^{l_2}}{l_1! l_2!} \mathfrak{T}_{(l_1,l_2)}(x) \right\}, \quad \alpha_i = n_i - 1, \quad x = \frac{z_{12}z_{34}}{z_{13}z_{24}}, \tag{1.12}$$

where $\mathfrak{T}_{(l_1,l_2)}(x)$ is defined by integrating connected stress-tensor correlators against entanglement temperatures of the associated intervals; see §5.3 for details. There are, however, subtleties in making sense of the formal expression due to divergence of $\mathfrak{T}_{(l_1,l_2)}(x)$ from singularities at coincident insertion points in stress-tensor correlators and the potential need to analytically continue in $\alpha_i$. We leave a more systematic study of the continuum modular Hamiltonian approach to future work.

**Discussions**

A few remarks are in order:

- In deriving our results, while we have relied on the formal manipulation of density matrices that technically don't exist in general in quantum field theory (see, e.g., [15] and references therein), our claims are mathematically precise and concern well-defined

mathematical objects such as isomonodromic tau functions.[3] It would be interesting to understand if the same results concerning TOC/tau functions can be derived without relying on the formal manipulations.

- We have focused on branched covers with monodromy data such that their associated TOCs can be identified as density matrix correlators involving disjoint intervals. More generic monodromy data can be obtained by taking the adjacent limit of the disjoint case, where the permutation monodromy in the limit is given by composition of monodromies at coincident endpoints. In fact, this implies that a generic genus-zero TOC/tau function can in principle be obtained from the class of multi-interval TOC/tau functions in (2.6) with $n_i = 2$; the reason is the following. As any permutation can be decomposed as composition of transpositions, generic genus-zero TOC/tau functions can be obtained by taking the coincident limit of higher-point genus-zero twist-two TOC/tau functions, i.e., those associated with *simple* branched covers. Furthermore, TOC/tau functions associated with branched covers with the same ramification profile are related by analytic continuation [12, 13], and therefore the desired twist-two TOC/tau functions can be obtained from the $n_i = 2$ case of (2.6) by continuation. Implementing this explicitly requires understanding better the regularization and monodromy property of the formal integral representation, which we leave for future work.

- The correlation matrix method for free fermions has been generalized to continuum in cyclic cases (i.e., Rényi entropies) in [16, 17] by generalizing the correlation matrix to an integral kernel and finding its spectrum and eigenfunctions. It would be interesting to understand if the non-abelian cases studied here can also be directly studied in the continuum using integral kernels.

- The relation between isomonodromic tau functions and CFT is also studied in [18–21], where the tau functions studied here are conjectured to be related to $W_N$ conformal blocks. To the best of our knowledge, the relation has not been made precise for the class of tau functions considered here due to technicalities in $W_N$ conformal blocks. The modular Hamiltonian approach for isomonodromic tau functions initiated here might also shed more light on the CFT/isomonodromy correspondence.

**Structure of the paper.** In §2, we discuss the general structures of TOC/tau functions and argue the approximate factorization property. In §3, we review the correlation matrix method for free fermions and derive the determinantal representation. In §4, we provide nontrivial checks of the determinantal representation and the approximate factorization property in several two-interval examples. In §5, we derive the formal integral representation and discuss some subtleties in the continuum modular Hamiltonian approach.

---

[3]Modulo the issue of showing the existence of the limits in the determinantal representations and making sense of the formal integrals in the integral representation. The existence of the limits are supported by direct numerical calculations in §4.

## Acknowledgments

This work is partially supported by funds from the University of California.

## 2 General structures and the approximate factorization property

In this section, we first discuss the general structures of TOC/tau functions and set up some notations used throughout the paper. We then argue an approximate factorization property of a class of TOC/tau functions utilizing known correlation structure of the ground state of large-$c$ holographic CFTs and the universality of genus-zero TOCs.

### 2.1 General structures of TOCs and tau functions

**Two intervals**

Recall the class of four-point TOCs mentioned in the introduction:

$$
\begin{aligned}
\mathcal{Z}_{(n_1,n_2)} &= \left|\tau_{(n_1,n_2)}\right|^{2c} = \mathrm{Tr}\left[\rho_R\left(\rho_{R_1}^{n_1-1} \otimes \rho_{R_2}^{n_2-1}\right)\right] = \left\langle \rho_{R_1}^{n_1-1}\rho_{R_2}^{n_2-1}\right\rangle \\
\sigma_1 &= \sigma_2^{-1} = (1\cdots n_1), \quad \sigma_3 = \sigma_4^{-1} = (1\ n_1+1\cdots n_1+n_2-1) \\
R &= R_1 \cup R_2, \quad R_i = (z_{2i-1}, z_{2i}).
\end{aligned} \tag{2.1}
$$

where $\langle\cdot\rangle$ denotes ground state expectation value.

The tau function can be thought of as the holomorphic part of the $c=1$ four-point genus zero TOC; in general, four-point functions in CFT only contain non-trivial dependence on cross-ratio:

$$
\begin{aligned}
\tau_{(n_1,n_2)}(\boldsymbol{z}) &= \mathcal{L}_{(n_1,n_2)}(\boldsymbol{z})\,\mathfrak{T}_{(n_1,n_2)}(x) := \prod_{i<j} z_{ij}^{\frac{h}{3}-h_i-h_j}\mathfrak{T}_{(n_1,n_2)}(x), \quad x = \frac{z_{12}z_{34}}{z_{13}z_{24}} \\
&= z_{12}^{-\frac{4}{3}\hat{h}_{n_1}+\frac{2}{3}\hat{h}_{n_2}}z_{34}^{\frac{2}{3}\hat{h}_{n_1}-\frac{4}{3}\hat{h}_{n_2}}\left(z_{13}z_{14}z_{23}z_{24}\right)^{-\frac{1}{3}\left(\hat{h}_{n_1}+\hat{h}_{n_2}\right)}\mathfrak{T}_{(n_1,n_2)}(x)
\end{aligned} \tag{2.2}
$$

where $h = \sum_i h_i = 2\hat{h}_{n_1} + 2\hat{h}_{n_2}$, and

$$
\hat{h}_n = \frac{1}{24}\left(n - n^{-1}\right) \tag{2.3}
$$

is the twist operator dimension with unit central charge. We denote the twist operator dimension for generic central charge by

$$
h_n = c\hat{h}_n. \tag{2.4}
$$

In some cases it is convenient to consider the tau function in special configuration $\boldsymbol{z} = (0, x, 1, \infty)$:

$$
\begin{aligned}
\tau_{(n_1,n_2)}(x) &:= \lim_{\boldsymbol{z}\to(0,x,1,\infty)} z_4^{2h_4}\tau_{(n_1,n_2)}(\boldsymbol{z}) \\
&= x^{-\frac{4}{3}\hat{h}_{n_1}+\frac{2}{3}\hat{h}_{n_2}}(1-x)^{-\frac{1}{3}\left(\hat{h}_{n_1}+\hat{h}_{n_2}\right)}\mathfrak{T}_{(n_1,n_2)}(x).
\end{aligned} \tag{2.5}
$$

**Multi-intervals**

We also consider the following more general class of genus-zero TOCs:

$$\mathcal{Z}_{(n_1,\cdots,n_r)} = \left|\tau_{(n_1,\cdots,n_r)}\right|^{2c} := \mathrm{Tr}\left[\rho_R\left(\bigotimes_{i=1}^{r}\rho_{R_i}^{n_i-1}\right)\right] = \left\langle\prod_{i=1}^{r}\rho_{R_i}^{n_i-1}\right\rangle$$

$$R = \bigcup_{i=1}^{r} R_i = \bigcup_{i=1}^{r}(a_i, b_i) = \bigcup_{i=1}^{r}(z_{2i-1}, z_{2i}). \tag{2.6}$$

The monodromies of the associated branched cover can be easily read off from the density matrix representation. In particular, the monodromies at endpoints of each interval $R_i$ are $n_i$ cycles, with different cycles for all the $r$ intervals having only one common sheet index (i.e., the one corresponding to $\rho_R$).

The $2r$-point tau function now depends on $2r - 3$ cross ratios:

$$\tau_{(n_1,\cdots n_r)}(\boldsymbol{z}) = \mathcal{L}_{(n_1,\cdots,n_r)}(\boldsymbol{z})\,\mathcal{T}_{(n_1,\cdots,n_r)}(\boldsymbol{x})$$

$$:= z_{ij}^{\delta_{ij}}\mathcal{T}_{(n_1,\cdots,n_r)}(\boldsymbol{x}) \tag{2.7}$$

with

$$\delta_{ij} = \frac{2}{2r-2}\left(\frac{\sum_k h_k}{2r-1} - h_i - h_j\right), \quad h_{2i-1} = h_{2i} = \hat{h}_{n_i}, \quad x_i = \frac{(z_i - z_{2r-2})(z_{2r-1} - z_{2r})}{(z_i - z_{2r-1})(z_{2r-2} - z_{2r})}. \tag{2.8}$$

## 2.2 Approximate factorization

**Ground state correlation structure in large-$c$ holographic CFTs**

In quantum information theory, the amount of correlation between two subsystems $R_1, R_2$ can be measured by the mutual information between them. The mutual information is defined as

$$I(R_1 : R_2) = S(R_1) + S(R_2) - S(R) = S_{\mathrm{rel}}(\rho_R|\rho_{R_1} \otimes \rho_{R_2}) \tag{2.9}$$

where $S(R)$ is the von Neumann entropy/entanglement entropy and $S_{\mathrm{rel}}$ the relative entropy. As the relative entropy is a distinguishability measure, the formulation of mutual information in terms of the relative entropy, i.e., as the distinguishability from a de-correlated state, makes manifest its information-theoretic meaning as a correlation measure. In particular,

$$I(R_1 : R_2) = 0 \iff \rho_R = \rho_{R_1} \otimes \rho_{R_2}, \tag{2.10}$$

i.e., mutual information vanishes iff the state factorizes.

In large-$c$ holographic CFTs, the mutual information in ground state $\rho_R$ between $R_1$ and $R_2$ is known to vanish to leading order in central charge in a specific kinematic regime:

$$I(R_1 : R_2) = \begin{cases} \mathcal{O}(c^0) & x \in \left(0, \frac{1}{2}\right) \\ \frac{c}{3}\log\left(\frac{x}{1-x}\right) & x \in \left(\frac{1}{2}, 1\right) \end{cases}, \tag{2.11}$$

with a phase transition at $x = \frac{1}{2}$. The statement can be derived using holographic entanglement entropy formula [22, 23] in $AdS_3/CFT_2$ or directly using large-$c$ CFT method [24]. This therefore implies that

$$\rho_R \simeq \rho_{R_1} \otimes \rho_{R_2}, \quad x \in \left(0, \frac{1}{2}\right) \quad \text{for large } c \text{ holographic CFT.} \tag{2.12}$$

**Implication for TOC/tau functions**

Now first consider the genus-zero TOCs $\mathcal{Z}_{(n_1,n_2)}$ in large-$c$ holographic CFT. In light of the density matrix interpretation of $\mathcal{Z}_{(n_1,n_2)}$ and the approximate factorization of $\rho_R$ in disconnected phase, we have

$$\mathcal{Z}_{(n_1,n_2)}(\boldsymbol{z}) \simeq \mathcal{Z}_{(n_1,n_2)}^{\text{fac.}}(\boldsymbol{z}), \quad x \in \left(0, \frac{1}{2}\right) \quad \text{for large } c \text{ holographic CFT}$$

$$\mathcal{Z}_{(n_1,n_2)}^{\text{fac.}}(\boldsymbol{z}) := \mathcal{Z}_{(n_1)}(z_1, z_2)\mathcal{Z}_{(n_2)}(z_3, z_4), \quad \mathcal{Z}_{(n)}(z_1, z_2) = |z_{12}|^{-4h_n}. \tag{2.13}$$

However, since the genus-zero TOCs $\mathcal{Z}_{(n_1,n_2)}$ have trivial dependence on central charge, we expect that the factorization approximation holds universally at the level of tau functions:

$$\tau_{(n_1,n_2)}(\boldsymbol{z}) \simeq \tau_{(n_1,n_2)}^{\text{fac.}}(\boldsymbol{z}), \quad x \in \left(0, \frac{1}{2}\right)$$

$$\tau_{(n_1,n_2)}^{\text{fac.}}(\boldsymbol{z}) := \tau_{(n_1)}(z_1, z_2)\tau_{(n_2)}(z_3, z_4), \quad \tau_{(n)}(z_1, z_2) = z_{12}^{-2\hat{h}_n}, \tag{2.14}$$

and furthermore that the restriction $x \in \left(0, \frac{1}{2}\right)$ can be relaxed as no phase transition is expected for tau functions. The factorized tau function $\tau_{(n_1,n_2)}^{\text{fac.}}$ corresponds to having the following cross-ratio dependent part:

$$\mathcal{T}_{(n_1,n_2)}^{\text{fac.}}(x) = \left(\frac{1-x}{x^2}\right)^{\frac{1}{3}\left(\hat{h}_{n_1} + \hat{h}_{n_2}\right)}. \tag{2.15}$$

We therefore have the following statement:

**Claim 2.1** (Approximate factorization of tau functions)**.** *There exists* $x^* \in \left(\frac{1}{2}, 1\right)$ *such that* $\forall x \in (0, x^*)$,

$$\left|\frac{\mathcal{T}_{(n_1,n_2)}(x)}{\mathcal{T}_{(n_1,n_2)}^{\text{fac.}}(x)}\right| = 1 + \epsilon, \quad |\epsilon| \ll 1. \tag{2.16}$$

**Remark 2.1.** In general, the TOC/tau functions satisfy the factorization limit

$$\lim_{x \to 0} \mathcal{T}_{(n_1,n_2)}(x) = \mathcal{T}_{(n_1,n_2)}^{\text{fac.}}(x), \tag{2.17}$$

which can be used to fix the normalization of $\mathcal{T}_{(n_1,n_2)}(x)$. That the factorization should approximately hold in a finite range of cross-ratio is a non-trivial statement.

# 3 TOCs and tau functions from correlation matrices

In this section, we derive determinantal representations for TOC/tau functions using the correlation matrix method for free fermions. The correlation matrix method is first developed in [14]; more details specific to discretization of free fermion CFT can be found in, e.g., [25].

## 3.1 Correlation matrix method for free fermion

Consider a theory of $\mathcal{N}$ fermions with algebra

$$\{\psi_i, \psi_j^\dagger\} = \delta_{ij}, \quad i = 1, \cdots, \mathcal{N} \tag{3.1}$$

and a quadratic Hamiltonian

$$\hat{H} = \sum_{i,j} \psi_i^\dagger M_{ij} \psi_j = \sum_i \lambda_i c_i^\dagger c_i, \quad \{c_i, c_j^\dagger\} = \delta_{ij} \tag{3.2}$$

where $\mathbf{M} = \mathbf{V}^\dagger \boldsymbol{\lambda} \mathbf{V}$ is a Hermitian matrix and $c_i = V_{ij}\psi_j, c_i^\dagger = \psi_j^\dagger \left(V^\dagger\right)_{ji}$.

The ground state is defined to be the state where negative energy modes are occupied and positive energy modes unoccupied:

$$c_i^\dagger |0\rangle = 0 \quad \text{for} \quad \lambda_i < 0, \qquad c_i |0\rangle = 0 \quad \text{for} \quad \lambda_i > 0. \tag{3.3}$$

Denote the ground state two-point function as

$$\mathsf{C}_{ij} = \left\langle \psi_i \psi_j^\dagger \right\rangle, \tag{3.4}$$

and it follows that

$$\mathsf{C} = \mathbf{V}^\dagger \theta(\boldsymbol{\lambda}) \mathbf{V} \tag{3.5}$$

where $\theta(\boldsymbol{\lambda})$ is diagonal with diagonal entries being ones for positive energy modes and zeros for negative energy modes.

We will be interested in the free fermion 2d CFT with Hamiltonian $-\frac{i}{2} \int dx \left(\psi^\dagger \partial \psi - \partial \psi^\dagger \psi\right)$, and therefore the discretized Hamiltonian

$$\hat{H} = -\frac{i}{2} \sum_j \psi_j^\dagger \psi_{j+1} - \psi_{j+1}^\dagger \psi_j$$

$$M_{jl} = -\frac{i}{2} \left(\delta_{l,j+1} - \delta_{l,j-1}\right). \tag{3.6}$$

The corresponding correlation matrix is given by

$$\mathsf{C}_{jl} = \begin{cases} \frac{(-1)^{j-l}-1}{2\pi i(j-l)} & j \neq l \\ \frac{1}{2} & j = l. \end{cases} \tag{3.7}$$

Let $R$ be a subset of the $\mathcal{N}$ indices, and define the correlation kernel by restriction to $R$:

$$(\mathsf{C}_R)_{ij} := \mathsf{C}_{ij}\big|_{i,j \in R}. \tag{3.8}$$

Let $\rho_R$ be the density matrix of $R$ with a quadratic modular Hamiltonian given by a modular Hamiltonian kernel $\mathsf{H}_R$:

$$\rho_R = \mathcal{N}_R^{-1} e^{-\mathcal{Q}(\mathsf{H}_R)}, \quad \mathcal{Q}(\mathsf{H}_R) := \sum_{i,j \in R} \psi_i^\dagger (\mathsf{H}_R)_{ij} \psi_j$$

$$\mathcal{N}_R = \operatorname{Tr} e^{-\mathcal{Q}(\mathsf{H}_R)} = \det\left(\mathbb{1} + e^{-\mathsf{H}_R}\right). \tag{3.9}$$

Requiring

$$(\mathsf{C}_R)_{ij} = \operatorname{Tr}\left(\rho_R \psi_i \psi_j^\dagger\right)\big|_{i,j \in R} \tag{3.10}$$

leads to following relation between modular Hamiltonian kernel and correlation kernel:

$$e^{-\mathsf{H}_R} = \mathsf{C}_R^{-1} - \mathbb{1}. \tag{3.11}$$

The relation allows one to write Rényi entropies directly in terms of correlation kernel. For example, the $N^{\text{th}}$ Rényi entropy can be written as:

$$\operatorname{Tr} \rho_R^N = \frac{\det\left(\mathbb{1} + e^{-N\mathsf{H}_R}\right)}{\det^N\left(\mathbb{1} + e^{-\mathsf{H}_R}\right)} = \det\left((\mathbb{1} - \mathsf{C}_R)^N + \mathsf{C}_R^N\right). \tag{3.12}$$

For more general density matrix correlators involving different density matrices, the computation is facilitated by the fact, as can be verified using free fermion algebra, that the quadratic modular Hamiltonian $\mathcal{Q}(\mathsf{H}_R)$ gives a representation of matrix algebra

$$\left[\mathcal{Q}\left(\mathsf{H}_R^{(1)}\right), \mathcal{Q}\left(\mathsf{H}_R^{(2)}\right)\right] = \mathcal{Q}\left(\left[\mathsf{H}_R^{(1)}, \mathsf{H}_R^{(2)}\right]\right), \tag{3.13}$$

from which it follows that

$$e^{-\mathcal{Q}\left(\mathsf{H}_R^{(1)}\right)} e^{-\mathcal{Q}\left(\mathsf{H}_R^{(2)}\right)} = e^{-\mathcal{Q}\left(\mathsf{H}_R^{(3)}\right)}$$

$$e^{-\mathsf{H}_R^{(1)}} e^{-\mathsf{H}_R^{(2)}} = e^{-\mathsf{H}_R^{(3)}}. \tag{3.14}$$

The property above is also used in other contexts such as [26, 27]. In the following, we compute the density matrix correlators in (2.1) by taking advantage of this property.

## 3.2 Relation with TOCs and tau functions

**Two intervals**

For the density matrix correlators associated with the class of four-point TOC/tau functions in (2.1), we have

$$\operatorname{Tr}\left[\rho_R \left(\rho_{R_1}^{n_1 - 1} \otimes \rho_{R_2}^{n_2 - 1}\right)\right]$$

$$= \det(\mathsf{C}_R) \det(\mathsf{C}_{R_1})^{n_1 - 1} \det(\mathsf{C}_{R_2})^{n_2 - 1} \operatorname{Tr}\left(e^{-\mathcal{Q}(\mathsf{H}_R)} e^{-\mathcal{Q}\left[(n_1 - 1)\mathsf{H}_{R_1} \oplus (n_2 - 1)\mathsf{H}_{R_2}\right]}\right)$$

$$= \det(\mathsf{C}_R) \det(\mathsf{C}_{R_1})^{n_1 - 1} \det(\mathsf{C}_{R_2})^{n_2 - 1} \det\left[\mathbb{1} + e^{-\mathsf{H}_R}\left(e^{-(n_1 - 1)\mathsf{H}_{R_1}} \oplus e^{-(n_2 - 1)\mathsf{H}_{R_2}}\right)\right]$$

$$= \det(\mathsf{C}_R) \det\left(\mathsf{C}_{R_1}^{n_1 - 1} \oplus \mathsf{C}_{R_2}^{n_2 - 1}\right) \det\left\{\mathbb{1} + (\mathsf{C}_R^{-1} - \mathbb{1})\left[\left(\mathsf{C}_{R_1}^{-1} - \mathbb{1}\right)^{n_1 - 1} \oplus \left(\mathsf{C}_{R_2}^{-1} - \mathbb{1}\right)^{n_2 - 1}\right]\right\}$$

$$= \det\left(\mathsf{M}_{R_1, R_2}^{(n_1, n_2)}\right), \tag{3.15}$$

with

$$\mathsf{M}^{(n_1,n_2)}_{R_1,R_2} := \mathsf{C}_R\left(\mathsf{C}^{n_1-1}_{R_1} \oplus \mathsf{C}^{n_2-1}_{R_2}\right) + \bar{\mathsf{C}}_R\left(\bar{\mathsf{C}}^{n_1-1}_{R_1} \oplus \bar{\mathsf{C}}^{n_2-1}_{R_2}\right), \quad \bar{\mathsf{C}} := \mathbb{1} - \mathsf{C}, \tag{3.16}$$

where we have used relations in (3.9), (3.11) and (3.14).

Now let $R_1, R_2$ be intervals with sizes $l_1, l_2$, respectively, and their separation being $d$. In terms of discrete indices on lattice, this corresponds to $R_1 = \{1, \cdots, l_1 + 1\}$, $R_2 = \{l_1 + d + 1, \cdots, l_1 + l_2 + d + 1\}$; the total number of fermions is $\mathcal{N} = l_1 + l_2 + d + 1$, and $\mathsf{M}^{(n_1,n_2)}_{R_1,R_2}$ is a $l_1 + l_2 + 2$ dimensional square-matrix. For this configuration, the leg factor and cross-ratio read:

$$\mathcal{L}_{(n_1,n_2)}(l_1, l_2, d) = l_1^{-\frac{4}{3}\hat{h}_{n_1}+\frac{2}{3}\hat{h}_{n_2}} l_2^{\frac{2}{3}\hat{h}_{n_1}-\frac{4}{3}\hat{h}_{n_2}}\left(d\,(l_1 + d)\,(l_2 + d)\,(l_1 + l_2 + d)\right)^{-\frac{1}{3}\left(\hat{h}_{n_1}+\hat{h}_{n_2}\right)},$$

$$x(l_1, l_2, d) = \frac{l_1 l_2}{(l_1 + d)(l_2 + d)}. \tag{3.17}$$

Due to the doubling of degree of freedom on lattice (cf., e.g., [25]), the density matrix correlators computed using correlation matrices above in fact correspond to $c = 1$ instead of $c = \frac{1}{2}$ TOCs. Recalling the general structure of TOC/tau functions discussed in §2.1, we can therefore extract the non-trivial part of TOC/tau functions $\mathcal{T}_{(n_1,n_2)}(x)$ as follows:

**Claim 3.1** (Determinantal representation of tau functions, two intervals). *The non-trivial part of the class of TOC/tau functions in* (2.1) *are related to the continuum limit of the determinant of free fermion correlation matrices by*

$$\left|\mathcal{T}_{(n_1,n_2)}(x)\right|^2 = \lim_{\substack{l_1,l_2,d\to\infty \\ x(l_1,l_2,d)\ \text{fixed}}} \mathcal{L}^{-2}_{(n_1,n_2)}(l_1, l_2, d) \det\left(\mathsf{M}^{(n_1,n_2)}_{R_1,R_2}\right) \tag{3.18}$$

*where the equality is understood to hold up to $x$-independent overall constant.*

**Multi-interval**

The generalization to the $2r$-point TOC and tau functions in (2.6) involving $r$-intervals is straightforward, and we simply state the result. We now consider $r$ intervals $R_i$ each with sizes $l_i$, with the separation between $R_i$ and $R_{i+1}$ being $d_i$. We can again translate the set-up to lattice similar to the two-interval case with $l_i, d_i$ being integers and $R_i$ being subsets of $\{1, \cdots \mathcal{N}\}$ with $\mathcal{N} = \sum_i l_i + \sum_i d_i + r - 1$. The associated leg factor $\mathcal{L}_{(n_1,\cdots,n_r)}(\boldsymbol{l}, \boldsymbol{d})$ and cross-ratios $\boldsymbol{x}(\boldsymbol{l}, \boldsymbol{d})$ can be easily determined from the general discussion in §2.1. Define the following $\sum_i l_i + r$-dimensional square matrix from correlation matrices:

$$\mathsf{M}^{(n_1,\cdots,n_r)}_{R_1,\cdots R_r} := \mathsf{C}_R\left(\bigoplus_{i=1}^{r} \mathsf{C}^{n_i-1}_{R_i}\right) + \bar{\mathsf{C}}_R\left(\bigoplus_{i=1}^{r} \bar{\mathsf{C}}^{n_i-1}_{R_i}\right). \tag{3.19}$$

The multi-interval generalization of our statement reads:

**Claim 3.2** (Determinantal representation of tau functions, multi-interval.). *The non-trivial part of the class of TOC/tau functions in (2.6) are related to the continuum limit of the determinant of free fermion correlation matrices by*

$$\left|\mathcal{T}_{(n_1,\cdots,n_r)}(\boldsymbol{x})\right|^2 = \lim_{\substack{\boldsymbol{l},\boldsymbol{d}\to\infty \\ \boldsymbol{x}(\boldsymbol{l},\boldsymbol{d}) \text{ fixed}}} \mathcal{L}_{(n_1,\cdots,n_r)}^{-2}(\boldsymbol{l},\boldsymbol{d}) \det\left(\mathsf{M}_{R_1,\cdots,R_r}^{(n_1,\cdots,n_r)}\right) \tag{3.20}$$

*where the equality is understood to hold up to $\boldsymbol{x}$-independent overall constant.*

## 4    Examples

In this section, we provide non-trivial checks of our claims 2.1 and 3.1 in several examples of two intervals. In the case of $n_1 = n_2 = 2$, the exact TOC/tau function is known analytically, and we will verify both claims 2.1 and 3.1 explicitly. For other examples of two intervals, we don't have the explicit knowledge of exact TOC/tau functions. Instead, in those examples, we will assume the validity of the claim 3.1 to verify the claim 2.1 by comparing $\mathcal{T}_{(n_1,n_2)}^{\text{fac.}}(x)$ with lattice data.

### 4.1    $n_1 = n_2 = 2$

The monodromy data in this case is given by

$$\sigma_1 = \sigma_2 = (12), \quad \sigma_3 = \sigma_4 = (13). \tag{4.1}$$

The associated tau function has known exact expression [20] in the special configuration $\boldsymbol{z} = (0, x, 1, \infty)$ (derivation reviewed in Appendix A):

$$\tau_{(2,2)}(x) = \frac{(3-\alpha)^{\frac{1}{3}}}{\sqrt{2}(1-\alpha)^{\frac{1}{8}}\left(\alpha(3+\alpha)\right)^{\frac{1}{24}}}, \quad x = \frac{(3+\alpha)^3(1-\alpha)}{(3-\alpha)^3(1+\alpha)}, \quad \alpha \in (0,1)$$

$$\mathcal{T}_{(2,2)}(x) = (x(1-x))^{\frac{1}{24}}\tau_{(2,2)}(x). \tag{4.2}$$

The factorization approximation is given by

$$\tau_{(2,2)}^{\text{fac.}}(x) = x^{-\frac{1}{8}}$$

$$\mathcal{T}_{(2,2)}^{\text{fac.}}(x) = (x(1-x))^{\frac{1}{24}}\tau_{(2,2)}^{\text{fac.}}(x) = \left(\frac{1-x}{x^2}\right)^{\frac{1}{24}}. \tag{4.3}$$

The normalization in $\mathcal{T}_{(2,2)}(x)$ was fixed by the factorization limit $\lim_{x\to 0}\mathcal{T}_{(2,2)}(x) = \mathcal{T}_{(2,2)}^{\text{fac.}}(x)$.

**Verification of claim 2.1.**    The correction to $\mathcal{T}_{(2,2)}^{\text{fac.}}(x)$ in the exact tau function $\mathcal{T}_{(2,2)}(x)$ is quite small for finite $x$; the small cross-ratio expansion of $\mathcal{T}_{(2,2)}(x)$ reads:

$$\frac{\mathcal{T}_{(2,2)}(x)}{\mathcal{T}_{(2,2)}^{\text{fac.}}(x)} = 1 + \frac{1}{2^9}x^2 + \frac{1}{2^9}x^3 + \frac{3^2 \cdot 103}{2^{19}}x^4 + \cdots \tag{4.4}$$

At $x = \frac{1}{2}$, the deviation from factorization approximation is

$$\frac{\mathcal{T}_{(2,2)}(\frac{1}{2})}{\mathcal{T}^{\text{fac.}}_{(2,2)}(\frac{1}{2})} \simeq 1 + 9.3 \times 10^{-4}. \tag{4.5}$$

The comparison between $\mathcal{T}_{(2,2)}(x)$ and $\mathcal{T}^{\text{fac.}}_{(2,2)}(x)$ in the full range of cross-ratio is given in Fig. 1. The deviation from factorization approximation increases with cross-ratio $x$, and the error is around 1% for $x \simeq .9$. This verifies the approximate factorization property in the claim 2.1 for the $n_1 = n_2 = 2$ case.

**Verification of claim 3.1.** The comparison between lattice data and $\mathcal{T}_{(2,2)}(x)$ is given in Fig. 2. The lattice data compute (the logarithm of) RHS of (3.18). Each data point with a particular cross-ratio $x$ is computed by choosing $l_1, l_2, d$ as described in §3.2 and increasing their sizes while keeping $x$ fixed until convergence is reached. An overall shift in $\log \mathcal{T}_{(2,2)}(x)$ is required to match the lattice data. We observe very good agreement between the exact expression for tau function $\mathcal{T}_{(2,2)}(x)$ and our lattice data. This verifies the determinantal representation of TOC/tau functions in claim 3.1 for the $n_1 = n_2 = 2$ case.

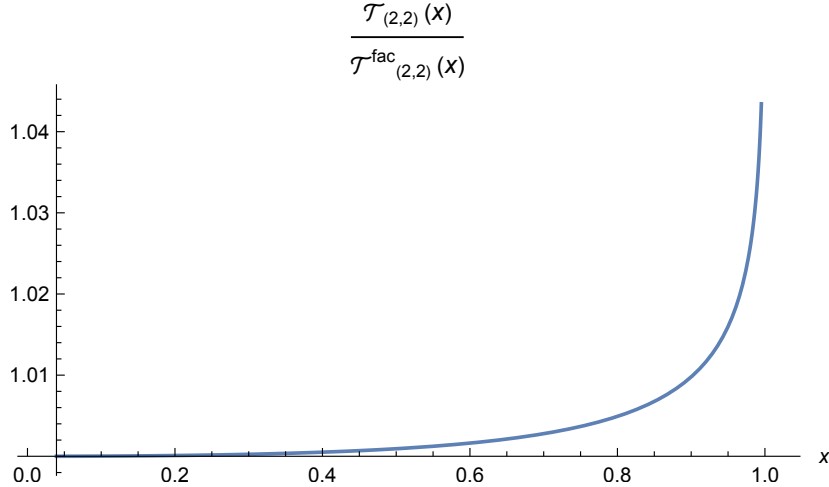

**Figure 1**: Comparison between $\mathcal{T}_{(2,2)}(x)$ and $\mathcal{T}^{\text{fac.}}_{(2,2)}(x)$. The deviation from factorization approximation increases with cross-ratio $x$, and the error is around 1% for $x \simeq .9$. This verifies the approximate factorization property in the claim 2.1 for the $n_1 = n_2 = 2$ case.

## 4.2 Other two-interval examples

Here we consider two other two-interval examples, with $(n_1, n_2) = (3, 2), (3, 3)$. As we don't know the exact TOC/tau functions in those cases, we will assume the validity of claim 3.1 that the lattice data compute the exact tau function $\mathcal{T}_{(n_1, n_2)}(x)$ and compare the factorized approximation $\mathcal{T}^{\text{fac.}}_{(n_1, n_2)}(x)$ with the lattice data. In the two examples, the factorized tau

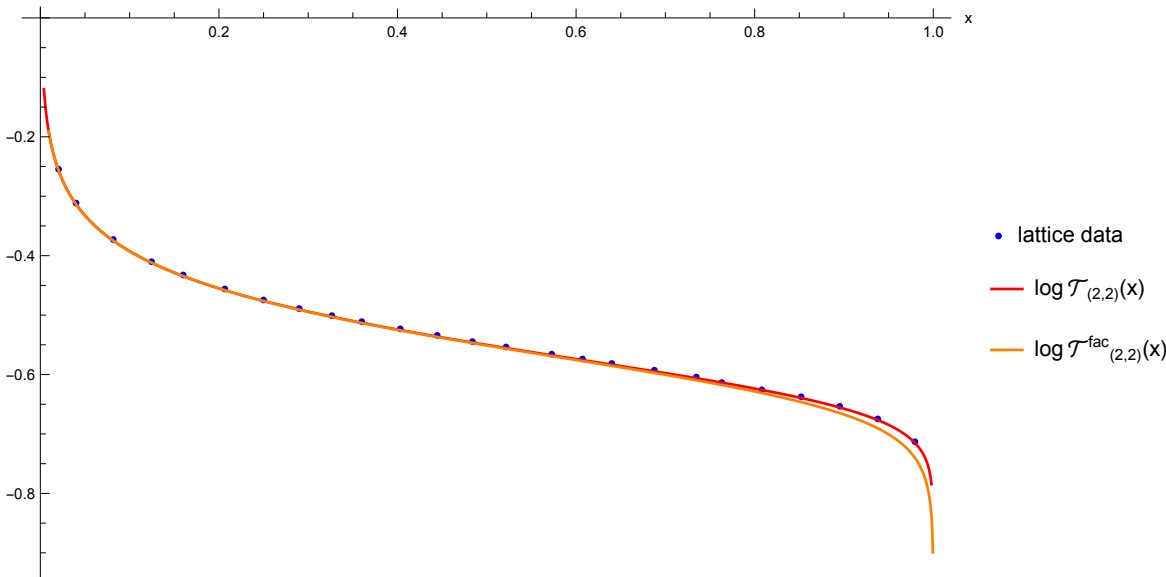

**Figure 2**: Comparison between lattice data and $\mathcal{T}_{(2,2)}(x)$. The lattice data compute (the logarithm of) RHS of (3.18). An overall shift in $\log \mathcal{T}_{(2,2)}(x)$ is required to match the lattice data. We observe very good agreement between the exact expression for tau function $\mathcal{T}_{(2,2)}(x)$ and our lattice data. This verifies the determinantal representation of TOC/tau functions in claim 3.1 for the $n_1 = n_2 = 2$ case.

functions read:

$$\mathcal{T}_{(3,2)}^{\text{fac.}}(x) = \left(\frac{1-x}{x^2}\right)^{\frac{25}{432}},$$

$$\mathcal{T}_{(3,3)}^{\text{fac.}}(x) = \left(\frac{1-x}{x^2}\right)^{\frac{2}{27}}. \tag{4.6}$$

The comparisons between the factorized TOC/tau functions and lattice data are given in Fig. 3. The lattice data are computed similarly to the case $(n_1, n_2) = (2, 2)$. The overall shifts in $\log \mathcal{T}_{(n_1,n_2)}^{\text{fac.}}(x)$ needed to compare with lattice data are determined by the factorization limit, i.e., by matching $\log \mathcal{T}_{(n_1,n_2)}^{\text{fac.}}(x)$ with lattice data at small cross-ratios. In both cases, we observe that, similar to the $(n_1, n_2) = (2, 2)$ case, the factorized tau functions give good approximation to lattice data in an extended range of cross ratio. This gives more evidence to the claim 2.1.

## 5 Towards a representation from continuum modular Hamiltonian

In this section, we study TOC/tau functions using the continuum expression for modular Hamiltonian of single interval [8–10]. This results in a formal integral representation in terms of integrated stress-tensor correlators for the class of TOC/tau functions in (2.1). We leave more careful study for certain subtleties/technicalities for future work.

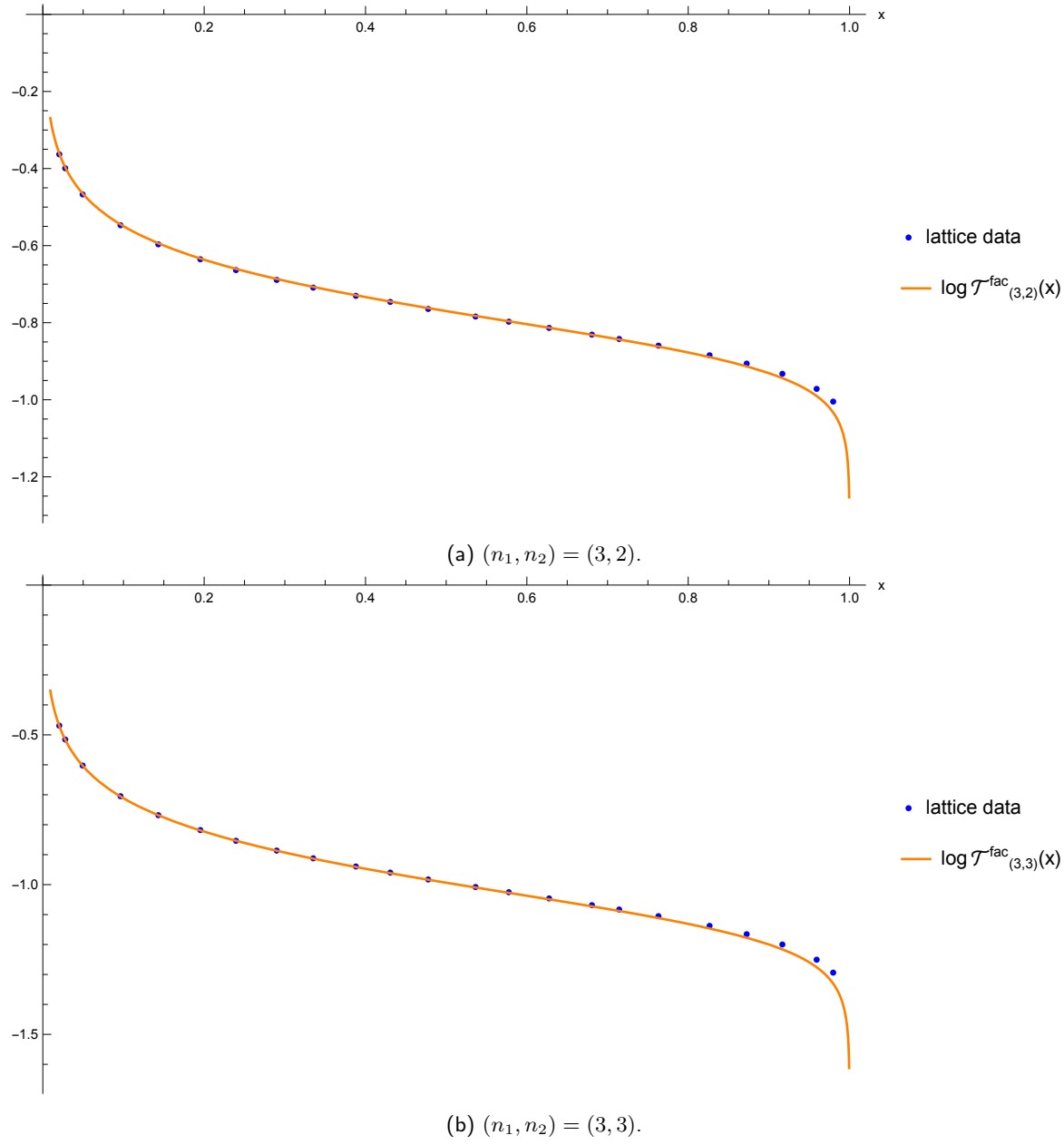

(a) $(n_1, n_2) = (3, 2)$.

(b) $(n_1, n_2) = (3, 3)$.

**Figure 3**: The comparisons between the factorized TOC/tau functions and lattice data in cases $(n_1, n_2) = (3, 2), (3, 3)$. The overall shifts in $\log \mathcal{T}^{\text{fac.}}_{(n_1, n_2)}(x)$ needed to compare with lattice data are determined by the factorization limit. In both cases, we observe that the factorized tau functions give good approximation to lattice data in an extended range of cross ratio. This gives more evidence to the claim 2.1.

**Note.** It is understood that the expressions for TOCs in this section are only their holomorphic parts.

## 5.1 Generalities

As mentioned in the introduction, the class of TOCs in (2.1) can in principle be computed from the universal expression for single-interval modular Hamiltonian $\mathcal{H}_{R_i}$:

$$\rho_{R_i} \propto e^{-\mathcal{H}_{R_i}}, \quad R_i = (a_i, b_i)$$

$$\mathcal{H}_{R_i} = \int_{R_i} dz \frac{(z - a_i)(z - b_i)}{b_i - a_i} T(z) + \frac{c}{6} \log \frac{b_i - a_i}{\epsilon} \mathbb{1} + \text{anti-holo..} \tag{5.1}$$

where the constant term in $\mathcal{H}_{R_i}$ is fixed by requiring the von Neumann entropy $S(R_i) = \langle \mathcal{H}_{R_i} \rangle$ has the known answer $S(R_i) = \frac{c}{3} \log \frac{b_i - a_i}{\epsilon}$. For convenience we write

$$\rho_{R_i} = (b_i - a_i)^{-\frac{c}{6}} e^{\mathcal{K}_{R_i}} \times \text{anti-holo.}$$

$$\mathcal{K}_{R_i} = \int_{R_i} dz \beta_i(z) T(z), \quad \beta_i(z) := \frac{(z - a_i)(z - b_i)}{a_i - b_i}. \tag{5.2}$$

The quantity $\beta_i(z)$ is sometimes referred to as "entanglement temperature" in literature.

The density matrix correlators in (2.1) can then be evaluated as sum of integrated stress-tensor correlators, which are fixed by the Ward identity:

$$\left\langle T(z) \prod_{i=1}^{n} T(z_i) \right\rangle = \left( \sum_i \frac{2}{(z - z_i)^2} + \frac{\partial_{z_i}}{z - z_i} \right) \left\langle \prod_{i=1}^{n} T(z_i) \right\rangle$$

$$+ \sum_i \langle T(z) T(z_i) \rangle \langle T(z_1) \cdots T(z_{i-1}) T(z_{i+1}) \cdots T(z_n) \rangle, \quad \langle T(z) T(z_i) \rangle = \frac{c/2}{(z - z_i)^4} \tag{5.3}$$

In other words, the Ward identity recursively defines higher-point stress-tensor correlators from the initial conditions of trivial one-point function $\langle T(z) \rangle = 0$ and two-point function $\langle T(z_1) T(z_2) \rangle = \frac{c/2}{z_{12}^4}$. The Ward identity guarantees that the recursively-defined stress-tensor correlators have the expected structure of splitting into connected part and disconnected part:

$$\left\langle \prod_i T(z_i) \right\rangle = \left\langle \prod_i T(z_i) \right\rangle_{\text{c}} + \left\langle \prod_i T(z_i) \right\rangle_{\text{disc}}, \tag{5.4}$$

with connected part given by the $\mathcal{O}(c)$ term:

$$\left\langle \prod_i T(z_i) \right\rangle_{\text{c}} = \left\langle \prod_i T(z_i) \right\rangle \Big|_{\mathcal{O}(c)}. \tag{5.5}$$

The connected, $\mathcal{O}(c)$ parts of the stress-tensor correlators can be recursively obtained from the usual weight-two, anomaly-free part of the Ward identity:

$$\left\langle T(z) \prod_{i=1}^{n} T(z_i) \right\rangle_{\text{c}} = \left( \sum_i \frac{2}{(z - z_i)^2} + \frac{\partial_{z_i}}{z - z_i} \right) \left\langle \prod_{i=1}^{n} T(z_i) \right\rangle_{\text{c}}. \tag{5.6}$$

**Diagrammatic notations.** We introduce following convenient diagrammatic notations for stress-tensor correlators:

$$\left\langle \prod_i T(z_i) \right\rangle = \text{\small •}\!\!\sim\!\!\sim\!\!\sim\!\!\text{•}\!\!\sim\!\!-\!\!\cdots\!\!\sim\!\!\text{•}$$

$$\left\langle \prod_i T(z_i) \right\rangle_{\mathrm{c}} = \text{•}\!-\!-\!-\!\text{•}\!-\!\cdots\!-\!\text{•} \quad , \tag{5.7}$$

and the non-trivial part of modular Hamiltonian $\mathcal{K}_{R_i}$:

$$\mathcal{K}_{R_i} = \int_{R_i} dz\, \beta_i(z) T(z) = \overset{i}{\underset{\bullet}{\big|}} \quad . \tag{5.8}$$

## 5.2 Single interval

As a consistency check of the method, first consider single interval $R = (a, b)$ case where the two-point TOC is known. In this case the TOC is given by:

$$\mathcal{Z}_{(n)} = \left\langle \rho_R^{n-1} \right\rangle = (b-a)^{-\frac{\alpha c}{6}} \left\langle e^{\alpha \mathcal{K}_R} \right\rangle, \quad \alpha = n - 1$$

$$\left\langle e^{\alpha \mathcal{K}_R} \right\rangle = \sum_l \frac{\alpha^l}{l!} \left\langle \mathcal{K}_R^l \right\rangle$$

$$= 1 + \sum_{l>1} \frac{\alpha^l}{l!} \underset{l}{\big\lceil\!\!\sim\!\!\sim\!\!\cdots\!\!\sim\!\!\sim\!\!\big\rceil}. \tag{5.9}$$

Consistency with the universal Weyl-anomaly structure of genus-zero TOCs requires the following exponentiation structure:

$$\left\langle e^{\alpha \mathcal{K}_R} \right\rangle = \exp\left( \left. \left\langle e^{\alpha \mathcal{K}_R} \right\rangle \right|_{\mathcal{O}(c)} \right), \tag{5.10}$$

or more explicitly,

$$1 + \sum_{l>1} \frac{\alpha^l}{l!} \underset{l}{\big\lceil\!\!\sim\!\!\sim\!\!\cdots\!\!\sim\!\!\big\rceil} = \exp\left( \sum_{l>1} \frac{\alpha^l}{l!} \underset{l}{\big\lceil\!-\!-\!-\!-\!\big\rceil} \right). \tag{5.11}$$

The multiplicities of distinct terms on LHS equal the number of partitions of a set with size $l$ with distinct partition structures (with partitions with size one excluded). The combinatorial structure is similar to the generating function of Bell number $B_n$ (the number of partitions of a set of size $n$):

$$\sum_n \frac{B_n}{n!} x^n = e^{e^x - 1}. \tag{5.12}$$

For convenience, define the following notation for integrated connected stress-tensor correlator

$$\mathfrak{K}_{(l)} = \left\langle \mathcal{K}_R^l \right\rangle \Big|_{\mathcal{O}(c)} = \overset{\rule{4cm}{0pt}}{\underset{l}{\rule{4cm}{0pt}}}\,,$$ (5.13)

the two-point TOC in the continuum modular Hamiltonian approach is therefore given by

$$\mathcal{Z}_{(n)} = (b-a)^{-\frac{\alpha c}{6}} \prod_{l>1} \exp\left( \frac{\alpha^l}{l!} \mathfrak{K}_{(l)} \right).$$ (5.14)

Comparing with known answer for two-point TOC

$$\mathcal{Z}_{(n)} = (b-a)^{-2h_n} = (b-a)^{-\frac{c}{12}\left(2\alpha - \alpha^2 + \alpha^3 - \alpha^4 + \cdots\right)},$$ (5.15)

implies that the integrated connected stress-tensor correlator should evaluate to

$$\mathfrak{K}_{(l)} = (-1)^l l! \frac{c}{12} \log(b-a)$$ (5.16)

where the equality is understood to hold up to cut-off in regularization of the associated integrals.

We verified by brute-force that the regularized integrals $\mathfrak{K}_{(l)}$ do agree with the expected expression above up to $l = 4$, and leave more systematic checks taking advantage of the recursion structure of stress-tensor correlators for future work. However, we observe the following subtlety: the expression (5.14) in fact would not converge for $\alpha \geq 1$ or $n \geq 2$. The correct answer for two-point TOC is only recovered when contributions from all $\mathfrak{K}_{(l)}$ are summed in (5.14) for $\alpha < 1$ and then analytically continued to $\alpha \geq 1$.

## 5.3 Two intervals

For the four-point TOCs in (2.1), we have

$$\mathcal{Z}_{(n_1,n_2)} = \left\langle \rho_{R_1}^{n_1-1} \rho_{R_2}^{n_2-1} \right\rangle = \left( \prod_{i=1}^2 (b_i - a_i)^{-\frac{\alpha_i c}{6}} \right) \left\langle e^{\alpha_1 \mathcal{K}_{R_1}} e^{\alpha_2 \mathcal{K}_{R_2}} \right\rangle, \quad \alpha_i = n_i - 1$$

$$\left\langle e^{\alpha_1 \mathcal{K}_{R_1}} e^{\alpha_2 \mathcal{K}_{R_2}} \right\rangle = \sum_{l_1, l_2} \frac{\alpha_1^{l_1} \alpha_2^{l_2}}{l_1! l_2!} \left\langle \mathcal{K}_{R_1}^{l_1} \mathcal{K}_{R_2}^{l_2} \right\rangle$$

$$= 1 + \sum_{l_1 + l_2 > 1} \frac{\alpha_1^{l_1} \alpha_2^{l_2}}{l_1! l_2!} \overset{\substack{1 \qquad 1 \quad 2 \qquad 2}}{\underset{\substack{l_1 \qquad\qquad l_2}}{\rule{6cm}{0pt}}}.$$ (5.17)

The universal Weyl anomaly structure of genus-zero TOC again requires

$$\left\langle e^{\alpha_1 \mathcal{K}_{R_1}} e^{\alpha_2 \mathcal{K}_{R_2}} \right\rangle = \exp\left( \left\langle e^{\alpha_1 \mathcal{K}_{R_1}} e^{\alpha_2 \mathcal{K}_{R_2}} \right\rangle \Big|_{\mathcal{O}(c)} \right),$$ (5.18)

i.e., the following combinatorial structure holds:

$$
1 + \sum_{l_1+l_2>1} \frac{\alpha_1^{l_1} \alpha_2^{l_2}}{l_1! l_2!} \quad \underset{l_1 \qquad\qquad l_2}{\overset{\displaystyle 1 \qquad\quad 1 \quad 2 \qquad\quad 2}{\vcenter{\hbox{\includegraphics}}}}
$$

$$
= \exp\left( \sum_{l_1+l_2>1} \frac{\alpha_1^{l_1} \alpha_2^{l_2}}{l_1! l_2!} \quad \underset{l_1 \qquad\qquad l_2}{\overset{\displaystyle 1 \qquad\quad 1 \quad 2 \qquad\quad 2}{\vcenter{\hbox{\includegraphics}}}} \right). \tag{5.19}
$$

For convenience, we now define, similar to the single interval case, the following notation for integrated connected stress-tensor correlators:

$$
\mathfrak{K}_{(l_1,l_2)} := \left\langle \mathcal{K}_{R_1}^{l_1} \mathcal{K}_{R_2}^{l_2} \right\rangle \Big|_{\mathcal{O}(c)} = \quad \underset{l_1 \qquad\qquad l_2}{\overset{\displaystyle 1 \qquad\quad 1 \quad 2 \qquad\quad 2}{\vcenter{\hbox{\includegraphics}}}}, \tag{5.20}
$$

and the four-point TOC in the continuum modular Hamiltonian approach is therefore given by:

$$
\mathcal{Z}_{(n_1,n_2)} = \left( \prod_{i=1}^{2} (b_i - a_i)^{-\frac{\alpha_i c}{6}} \right) \prod_{l_1+l_2>1} \exp\left\{ \frac{\alpha_1^{l_1} \alpha_2^{l_2}}{l_1! l_2!} \mathfrak{K}_{(l_1,l_2)} \right\}. \tag{5.21}
$$

Interestingly, the resulting expression from the continuum modular Hamiltonian approach manifestly splits into the factorized answer and its correction: the $l_1 = 0$ and $l_2 = 0$ parts of the previous expression give $\mathcal{Z}_{(n_1,n_2)}^{\text{fac.}} := \mathcal{Z}_{(n_1)} \mathcal{Z}_{(n_2)}$. We therefore have, for the class of four-point TOCs in (2.1):

$$
\frac{\mathcal{Z}_{(n_1,n_2)}(\boldsymbol{z})}{\mathcal{Z}_{(n_1,n_2)}^{\text{fac.}}(\boldsymbol{z})} = \prod_{\substack{l_1+l_2>1 \\ l_1,l_2\neq0}} \exp\left\{ \frac{\alpha_1^{l_1} \alpha_2^{l_2}}{l_1! l_2!} \mathfrak{K}_{(l_1,l_2)}(x) \right\}, \quad \alpha_i = n_i - 1, \quad x = \frac{z_{12}z_{34}}{z_{13}z_{24}}. \tag{5.22}
$$

To make contact with our previous tau function notations, we define the $c = 1$ version of the integrated stress-tensor correlators:

$$
\mathfrak{T}_{(l_1,l_2)} := \mathfrak{K}_{(l_1,l_2)} \big|_{c=1}, \tag{5.23}
$$

and we have (the common leg factor has been canceled below):

$$
\frac{\mathfrak{T}_{(n_1,n_2)}(x)}{\mathfrak{T}_{(n_1,n_2)}^{\text{fac.}}(x)} = \prod_{\substack{l_1+l_2>1 \\ l_1,l_2\neq0}} \exp\left\{ \frac{\alpha_1^{l_1} \alpha_2^{l_2}}{l_1! l_2!} \mathfrak{T}_{(l_1,l_2)}(x) \right\}, \quad \alpha_i = n_i - 1, \quad x = \frac{z_{12}z_{34}}{z_{13}z_{24}}. \tag{5.24}
$$

In particular, our claim 2.1 states that the (appropriately regularized) RHS of previous expression should be very close to one in a finite range of cross-ratio.

We have the following symmetry property for $\mathfrak{T}_{(l_1,l_2)}(x)$ due to its dependence on cross-ratio:

$$\mathfrak{T}_{(l_1,l_2)}(x) = \mathfrak{T}_{(l_2,l_1)}(x), \tag{5.25}$$

as the cross-ratio is unchanged after exchanging the two intervals.

We warn the reader that the integral representations for TOC/tau functions in preceding expressions involving $\mathfrak{K}_{(l_1,l_2)}$ are formal due to singularities in stress-tensor correlators at coincident insertion points, and the associated integrals need to be properly regularized. For explicit evaluation, it is enough to consider special configuration $R_1 = (0,x), R_2 = (1,\infty)$. A convenient change of variable yields:

$$\mathfrak{K}_{(l_1,l_2)}(x) = \int\limits_{[0,1]^{(l_1+l_2)}} \left[\prod_{i=1}^{l_1} \hat{\beta}_1\left(\mathbf{z}_i\right) d\mathbf{z}_i\right] \left[\prod_{j=1}^{l_2} \hat{\beta}_2\left(\mathbf{w}_j\right) d\mathbf{w}_j\right] \left\langle \prod_{i=1}^{l_1} T\left(x\mathbf{z}_i\right) \prod_{j=1}^{l_2} T\left(\frac{1}{\mathbf{w}_j}\right) \right\rangle_{\mathrm{c}}$$

$$\hat{\beta}_1\left(\mathbf{z}\right) = x^2 \mathbf{z}(1-\mathbf{z}), \quad \hat{\beta}_2\left(\mathbf{w}\right) = \frac{1-\mathbf{w}}{\mathbf{w}^3}. \tag{5.26}$$

For example,

$$\mathfrak{K}_{(1,1)}(x) = \frac{cx^2}{2} \int\limits_{[0,1]^2} d\mathbf{z} d\mathbf{w} \frac{\mathbf{z}\left(1-\mathbf{z}\right) \mathbf{w}\left(1-\mathbf{w}\right)}{\left(1-x\mathbf{z}\mathbf{w}\right)^4}$$

$$= -\frac{c}{12}\left[2 + \left(\frac{2}{x}-1\right)\log\left(1-x\right)\right], \tag{5.27}$$

$$\mathfrak{K}_{(2,1)}(x) = cx^2 \, \mathrm{reg.} \int\limits_{[0,1]^3} d\mathbf{z}_1 d\mathbf{z}_2 d\mathbf{w} \frac{\mathbf{z}_1\left(1-\mathbf{z}_1\right) \mathbf{z}_2\left(1-\mathbf{z}_2\right) \mathbf{w}\left(1-\mathbf{w}\right)}{\left(\mathbf{z}_1-\mathbf{z}_2\right)^2 \left(1-x\mathbf{z}_1\mathbf{w}\right)^2 \left(1-x\mathbf{z}_2\mathbf{w}\right)^2}$$

$$= \frac{c}{4}\left[2 + \left(\frac{2}{x}-1\right)\log\left(1-x\right)\right], \tag{5.28}$$

where $\mathfrak{K}_{(2,1)}$ is technically divergent, and we have regularized by imposing cut-offs at endpoints. The quantity $\mathfrak{K}_{(l_1,l_2)}$ at small $l_i$ is also studied in [28].

As noted in [28], brute-force evaluation of $\mathfrak{K}_{(l_1,l_2)}$ quickly becomes cumbersome with increasing $l_i$, and we leave for future work to systematically study their properties such as proper regularization and efficient evaluation. Moreover, another concern is the convergence in the sum over $l_i$ and whether continuation in $\alpha_i$ is required as in the single interval case. We also leave a careful study of the issue to future work.

It is clear that the continuum modular Hamiltonian approach can be also generalized to the multi-interval set-up (2.6), and that similar issues regarding regularization arise. We therefore also leave a more explicit discussion of the multi-interval case for future work.

## A   Derivation of (4.2)

Here we review the derivation of (4.2) in [20], where a one-parameter family of genus zero branched covers

$$\phi : \mathbb{CP}^1 \to \mathbb{CP}^1$$
$$w \mapsto z \tag{A.1}$$

with monodromy data

$$\mathbf{z} = (0, x(\alpha), 1, \infty), \quad \alpha \in (0,1)$$
$$\sigma_1 = \sigma_2 = (12), \quad \sigma_3 = \sigma_4 = (13) \tag{A.2}$$

is constructed, with

$$\phi(w) = \frac{\left(2w - \alpha^2 + 1\right)^2 (w - 4)}{(\alpha - 3)^2 (\alpha + 1)w}$$
$$x(\alpha) = \frac{(3 + \alpha)^3 (1 - \alpha)}{(3 - \alpha)^3 (1 + \alpha)}. \tag{A.3}$$

We will need the pre-images of the branched point $x(\alpha)$,

$$\phi^{-1}(x(\alpha)) = \{w_2^{\text{br.}}, w_2^{\text{non.br.}}\} = \{1 - \alpha, (1 + \alpha)^2\}. \tag{A.4}$$

Among the two pre-images, one of them is branched corresponding to the two-cycle, and the other is not branched corresponding to the trivial cycle.

The associated tau function is given by (below $\psi^I$ denotes inverses of $\phi$, and residues are understood as residues of one-forms)

$$\partial_x \log \tau_{(2,2)} = \frac{1}{12} \text{Res}_{z=x} \sum_I \{\psi^I, z\} dz$$

$$= -\frac{1}{12} \text{Res}_{z=x} \sum_I \left(\frac{d\psi^I}{dz}\right)^2 \{\phi, w\}\big|_{w=\psi^I(z)} dz$$

$$= -\frac{1}{12} \left(\text{Res}_{w=w_2^{\text{br.}}} + \text{Res}_{w=w_2^{\text{non.br.}}}\right) \frac{\{\phi, w\}}{\phi'(w)} dw$$

$$= -\frac{1}{12} \text{Res}_{w=w_2^{\text{br.}}} \frac{\{\phi, w\}}{\phi'(w)} dw$$

$$= -\frac{(\alpha - 3)^3 (\alpha + 1)^3 \left(\alpha^2 + 6\alpha - 3\right)}{128(\alpha - 1)\alpha^3 (\alpha + 3)^3} \equiv f(\alpha) \tag{A.5}$$

Solving the equation

$$\partial_\alpha \log \tau_{(2,2)} = x'(\alpha) f(\alpha) \tag{A.6}$$

yields

$$\tau_{(2,2)} \propto \frac{(3 - \alpha)^{\frac{1}{3}}}{(1 - \alpha)^{\frac{1}{8}} \left(\alpha (3 + \alpha)\right)^{\frac{1}{24}}}. \tag{A.7}$$

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
