# Peer review of "Twist operator correlators and isomonodromic tau functions from modular Hamiltonians"

_SciPost Physics_

## Round 2 · Referee Report · Anonymous (Referee 1) · 2024-7-31

Strengths

result interesting
subject of current interest

Weaknesses

proofs are too difficult to read and fully assess

Report

The paper under review is a continuation of the previous paper https://arxiv.org/pdf/2307.03729. by the same author; therefore some of the comments are applicable to both of these works.

The main goal of this work is to relate certain correlators of CFT's defined on a branched cover of the Riemann sphere to isomonodromic tau-functions of special Riemann-Hilbert problems with quasi-permutation monodromy matrices. the author restricts himself to coverings of genera 0 and 1, when such connection can be made explicit.

The relation between CFT and isomonodromy equations is an interesting topic which attracted a lot of attention recently (the Kyiv formula by Gamayun, Iorgov and Lisovyy, the works and more recent papers by Gavrylenko-Marshakov).

The correlator of twist (or permutation) operators on the Riemann sphere defined by the path integral (1.1) can be alternatively understood as a partition function of the CFT defined on the corresponding branched covering which for the cases of free bosons or free fermions was studied in several classical papers in 1980's (Knizhnik, Sonoda, Alvarez etc). The author considers the case of an arbitrary central charge and considers this object from the perspective of Calabrese-Cardy work [2] where cyclic coverings of genus 0 were considered.

Using the defining equations for the twist correlators the author relates them to the Bergman tau-function which is one of the components of the isomonodromic tau-function of the Riemann-Hilbert problem with quasi-permutation monodromy matrices.

The main results, Claims 3.1 and 3.2, look new and interesting, although I was unable to get through the technicalities of the proofs. The author checks a few special cases and confirms his formulas via comparison with lattice approximation.

The results seem reasonable, although, as I stated above, I can not verify all the details of the proof, partially due to the use of physics jargon in the proof of pure mathematical statements.

Assuming the proofs can be written in more detail the paper can become suitable for publication.

I can offer a few simple comments:

paper 2307.03729:

  1. the use of the letter tau for the period of the curve and the tau-function simultaneously needs to modified.

  2. In (2.3) the author needs to mention that $w$ is the normalized holomorphic differential on the torus (the a-period is 1). The freedom of the choice of the a-cycle should be discussed in the context of the author's approach (the stress-energy tensor does depend on this choice).

  3. In (2.10) the normailization of the basis \omega_\alpha should be stated. It should be added that these are holomorphic differentials.

  4. (2.31): Only the residue of a 1-form makes sense. The author has the residue of the projective connection in this formula which does not have a mathematical meaning. What should replace this is a difference of two projective connections (to get a quadratic differential) divided by some 1-form. The same in (2.32).

  5. The same in (2.36): the residue of quadratic differential does not make sense; this formula should written properly in a coordinate-independent way.

  6. If the path integral (1.1) can be defined as a partition function of CFT on the branch covering (with some conditions on behaviour of all fields at the brach points and infinities of each sheet) it should be clearly explained.

  7. I presume that N_{g,N} in (1.5) is the Hurwitz number; if this is the case it should be clearly stated.

  8. It would be desirable to discuss in detail the case of genus 0ne formula (3.40) in the case of free bosons. Does Z_m have the meaning of the determinant of laplacian on the branch cover in the singular metric lifted from the base?

  9. Notations in (3.47) need to be changes again: there are 3 different \tau in this formula! The essential question is: the bosonic partition function is the Bergman tau-function of [17]; it eneters the tau-function of the quasi-permutation monodromy group with power -1/2, not 1.

the above comments propagate to the paper 2308.16839:

  1. use of residue in (1.2), (1.8) etc
  2. correct power of \tau_m in (1.3)?
  3. what monodromy data the author is talking about after (1.6)?
  4. page 2, second paragraph: what does it mean to "explicitly construct the brach cover"? The author fixes the branch points and permutations everywhere which gives the branch cover by definition.
  5. The tau-function in left hand side of (1.10) is known explicitly via resultants and theta-functions (in genus 1). What is the advantage of the limit representation we have in the r.h.s. here?
  6. (1.11): Dies x^* depend on the choice of \epsilon here?
  7. The author over-uses capital greek notations; it makes the paper hard to read.
  8. The computation is sec.5.3 seems to be done for genus zero covers with 4 brach points. That needs to be explained in detail; it was hard for me to go through this proof.

Requested changes

see the report

Recommendation

Ask for major revision

---

## Editorial Decision

awaiting_resubmission